# *Know Yourself and Know Your Neighbour*: A Syntactically Informed Self-Supervised Compositional Sentence Representation Learning Framework using a Recursive Hypernetwork

**Vasudevan Nedumpozhimana**                                    *vnedumpo@tcd.ie*
**John Kelleher**                                          *john.kelleher@tcd.ie*
*ADAPT Research Centre*
*Trinity College Dublin, Ireland*

**Reviewed on OpenReview:** *https://openreview.net/forum?id=gfBiJv7r51*

## Abstract

Sentence representation learning is still an open challenge in Natural Language Processing. In this work, we propose a new self-supervised framework for learning sentence representations, using a special type of neural network called a recursive hypernetwork. Our proposed model composes the representation of a sentence from representations of words by applying a recursive composition through the parse tree. We maintain a separate syntactic and semantic representation, and the semantic composition is guided by the information from the syntactic representation. To train this model, we introduce a novel set of six self-supervised tasks. By analysing the performance on 7 probing tasks, we validate that the generated sentence representation encodes richer linguistic information than both averaging baselines and state-of-the-art alternatives. Furthermore, we assess the impact of the six proposed self-supervised training tasks through ablation studies. We also demonstrate that the representations generated by our model are stable for sentences of varying length and that the semantic composition operators adapt to different syntactic categories[1].

## 1 Introduction

Most neural language models generate vector representations (embeddings) of words or tokens. For example, earlier models such as word2vec (Mikolov et al., 2013) or GloVe (Pennington et al., 2014) generate representations of words, and more recent models such as BERT (Devlin et al., 2018) and RoBERTa (Liu et al., 2019) generate contextual representations of each token in a sentence. Most recent large language models, such as GPT (Brown et al., 2020) and Llama (Touvron et al., 2023), also internally generate representations of the tokens in the prompt for text generation. However, many natural language processing tasks require representations of sentences.

The standard way to create a sentence representation is to average the embeddings of the tokens in the sentence. However, this averaging ignores syntactic information and the position of tokens in a sentence. Although Transformer-based language models internally encode some syntactic information (Raganato & Tiedemann, 2018; Hewitt & Manning, 2019; Clark et al., 2019; Reif et al., 2019; Jawahar et al., 2019; Lin et al., 2019; Manning et al., 2020; Arps et al., 2022; Pimentel et al., 2022), such internal syntactic information is limited (Sinha et al., 2021b; Pham et al., 2021; Sinha et al., 2021a; Nedumpozhimana & Kelleher, 2024), and there is a growing body of work across a range of NLP tasks that points to the benefits of explicitly injecting syntactic information into transformer models (Pang et al., 2019; Moradshahi et al., 2019; Min et al., 2020; Wang et al., 2020; Pham et al., 2021; Sachan et al., 2021; Xu et al., 2021b; Tian et al., 2022; Hou et al., 2024; Kai et al., 2024).

---

[1]Code available: https://github.com/vasudev2020/codi/tree/main

Muennighoff et al. (2023) introduced the Massive Text Embedding (MTEB) benchmark and evaluated a range of embedding models across a variety of tasks. Their results indicate that no single embedding method obtains the best performance across all tasks, and when performance is averaged across tasks, supervised methods, in which at least one supervised task (such as NLI) with a labelled dataset is used to train the sentence representation model, outperform pure self-supervised methods. However, such supervised methods also have larger variance in performance across tasks. It has long been known that in deep networks the features learned in later layers are fitted to the training task and less transferable across tasks, see e.g. Yosinski et al. (2014). Consequently, the greater variance in performance across tasks for supervised methods may be caused by their representations being overfitted to the learning task, e.g., NLI task. Indeed, Carlsson et al. (2021) argue that self-supervised methods for sentence representations are preferable to supervised because they avoid the pre-training objective bias, and do not require labelled datasets.

Murty et al. (2023) observe that Transformer-based models trained in a supervised manner on compositional generalisation benchmarks become more tree-like (syntactic) in their processing over the course of training, and the more tree-like the processing of a model, the better its performance on tests of compositional generalizability. Moreover, they find that models trained in a self-supervised manner do not learn tree-structured processing. This suggests that one way to improve the performance of self-supervised methods for generating sentence representations is to improve the encoding of syntax within the generated embeddings.

We propose a novel self-supervised framework for generating syntactically informed sentence representations using a recursive hypernetwork. This framework has two major components: (1) a recursive hypernetwork architecture that implements a pair of trainable compositional operators, one semantic and one syntactic, and (2) a set of 6 self-supervised learning tasks used to train the compositional operators. The framework is designed to explicitly infuse syntactic information in two ways. First, the order in which embeddings are composed is defined by the constituency parse tree of the sentence being processed. Second, the 6 self-supervised training tasks were designed to guide the compositional operators to encode syntactic information as well as distributional semantic information in the embeddings they generate; we analyse the contribution of each of these tasks to the overall performance of the system in Section 5.1.

The rest of the paper is structured as follows. Section 2 presents a review of previous work, Section 3 describes the framework, and Section 4 describes the training of the models. Section 5 presents results from a set of probing tasks designed to assess whether the framework improves the encoding of linguistic information when compared with state-of-the-art baselines, and Section 6 reports an analysis of semantic compositional operator in terms of: (a) robustness to parse tree depth, and (b) sensitivity to syntactic information. Finally, Section 7 presents our conclusions and plans for future work.

## 2 Sentence Representation Learning

There is a growing body of work on generating neural sentence representations; see Kashyap et al. (2023) for a recent survey. The technical novelty of our work arises from combining a recursive hypernetwork with a self-supervised learning objective. Recursive hypernetworks have previously been used to train sentence representations; however, all of this work has been done using supervised learning. In the following two sections, we first review previous research on recursive networks for sentence representations and then review supervised and self-supervised approaches to sentence representation learning. Although to date supervised approaches tend to outperform unsupervised methods, this may be because unsupervised methods do not explicitly integrate syntactic information into their objectives (see discussion regarding Murty et al. (2023) in Section 1). Consequently, we hypothesise that a self-supervised framework (recursive hypernetwork model+objective) that encourages the encoding of more linguistic information into a representation may result in improved performance. The benefit of improving the performance of self-supervised methods is that it avoids the need for expensive labelled datasets, and may also result in more transferable (as in downstream task-agnostic) representations.

## 2.1 Recursive Neural Networks

Recursive Neural Networks are a deep learning architecture used to process hierarchical input, particularly the tree-structured input. These networks apply the same set of weights recursively over the structured input in topological order, from leaf to root in the case of a tree. In our work, we choose to use a recursive neural network for sentence representation learning because of its ability to explicitly use the syntactic information from a parse tree by composing the representation of sentences (or phrases) recursively[2]. Socher et al. (2011b;a) proposed a recursive neural network with a simple feedforward network as the composition operator, which was later generalised and extended by Socher et al. (2012; 2013a;b) for sentence representation learning. However, these recursive neural networks based on feedforward networks suffer from the vanishing gradient problem when processing sentences with deep tree structures. To address this problem, Tai et al. (2015) adapted the LSTM architecture to recursive networks and proposed a compositional operator called Tree-LSTM. Contemporaneously, and in parallel, Le & Zuidema (2015) and Zhu et al. (2015) proposed similar extensions that use an LSTM over a tree structure.

Kim et al. (2018) modified the Tree-LSTM by augmenting the syntactic tag information as a supplementary input to the gate functions of the Tree-LSTM to compose the representation of each node in the constituency parse tree. The traditional Tree-LSTM is designed to be applied on a binarised constituency parse tree, which can be one of its limitations. To overcome this, Xu et al. (2021a) extended the Tree-LSTM to an ARTree-LSTM, which can be applied to constituency parse trees with any number of child nodes. Liu et al. (2017) extended the Tree-LSTM by exploiting the advantages of hypernetworks or meta-networks. They used a small meta-network and a dynamic parameter prediction method to enable different types of compositions. Shen et al. (2020) extended this work further by explicitly using syntactic (tag) information for the dynamic parameter prediction and improved the performance. Xu et al. (2020) also used a similar combination of Tree-LSTM and hypernetwork for the compositional operator as that of Shen et al. (2020).

In the literature, the recursive neural networks are used to generate task-specific sentence representations. In our work, we use the combination of the hypernetwork with the Tree-LSTM (please see section 3 for more details) to compose sentence representations that avoid the need for separate training for each downstream task. To our knowledge, no one has explored the possibility of using the state-of-the-art recursive neural network (hypernetwork+Tree-LSTM) for composing such a general sentence representation.

## 2.2 Supervised versus Unsupervised

Some of the earlier models, such as by Hill et al. (2016); Mitchell & Lapata (2010); Mikolov et al. (2013); Arora et al. (2017), use simple non-parametric compositions for generating sentence representation. Apart from such simple baseline models, almost all approaches use some task(s) to learn or fine-tune the model to generate sentence representation, and Hill et al. (2016) observed that such tasks will significantly impact the quality of the model.

A considerable amount of literature has used supervised learning approaches to generate sentence representations specific to target downstream tasks. Such models have to train separately for each of the tasks, which is a limitation of this approach. Therefore, in this work, our focus is on generating representations of sentences that are not fitted to specific downstream tasks. There are some notable models in the literature that utilise the supervised learning approach for generating such sentence representations. For example, Conneau et al. (2017) proposed a supervised model called Infersent for learning universal sentence representation by using labelled Stanford Natural Language Inference (SNLI) datasets, and Wieting et al. (2016) proposed a model which captures paraphrastic similarity by using a paraphrase dataset for learning sentence representation. The Universal Sentence Encoder by Cer et al. (2018) utilised unlabeled data along with the labelled SNLI data to train a Transformer encoder model for generating sentence representation with a multitask learning framework. Reimers & Gurevych (2019) proposed another supervised model called SBERT, which is finetuned by using the SNLI labelled dataset after a self-supervised pre-training. Even though these supervised learning tasks empirically show their transfer effect to other tasks, there is no convincing reason

---

[2]Note that Recursive Neural Networks are not the only model that explicitly utilises the syntactic information from the parse tree. For example, the hypertree neural network proposed by Zhou et al. (2022) learns representations of each node in the constituency parse tree through iterative updates rather than a recursive composition.

observed for their generalizability (Carlsson et al., 2021). Also, such a supervised approach requires highly valuable labelled data, which restricts the scalability of such models. Therefore, generally, self-supervised or unsupervised tasks are preferred over supervised tasks for learning scalable models.

Self-supervised learning is widely used for generating sentence representations. One such model is the sentence representation model proposed by Le & Mikolov (2014), which is trained to predict words in the sentence (or document). The CPHRASE model by Pham et al. (2015) learns the representation by using the task of context predictions for phrases at all levels of the constituency parse tree generated by a supervised parser. The Skipthought model, Kiros et al. (2015) extended the skip-gram model for words (Mikolov et al., 2013) to the sentence level by proposing a task of decoding the next and previous sentences from the encoded sentence representation by using the encoder-decoder architecture. Ba et al. (2016) used the same task and architecture of Skipthought with an additional layer-norm regularisation and improved the performance. Hill et al. (2016) used an additional autoencoder task along with the task used by Kiros et al. (2015). In the same work, Hill et al. (2016) further proposed a Sequential Denoising Autoencoder model for sentence representation learning that has the objective of reconstructing the original sentence from a corrupted (noise-added) input representation.

Logeswaran & Lee (2018) simplified the tasks of Kiros et al. (2015) and used the task of identifying a context sentence from other contrastive sentences using the sentence representation, rather than regenerating the sentence itself. Pagliardini et al. (2018) used a CBOW-like objective (Mikolov et al., 2013) to predict a word within the sentence for composing the sentence representation. In another model called DeCLUTR, based on the concept of contrastive loss, Giorgi et al. (2021) extended the task of Logeswaran & Lee (2018) by allowing overlapping and subsuming context segments along with adjacent segments as positive samples. Carlsson et al. (2021) also proposed a self-supervised training objective called contrastive tension to learn sentence representation from pre-trained language models by removing the bias posed by the pre-training objective and achieved a new state-of-the-art in semantic text similarity tasks. Hu et al. (2022) used another self-supervised task of token prediction by using its left context and right context.

Inspired by various approaches in the literature, in this work, we propose a set of novel self-supervised tasks for learning representations of phrases and sentences, which we will describe in more detail in section 3.1. Some of our proposed tasks have some similarities with the tasks proposed or used by Pham et al. (2015) and Kiros et al. (2015). However, our tasks are novel and have many notable differences from previous tasks, and we will discuss more about this in section 3.1.

## 3 Proposed Model

The proposed model (shown in Figure 1) consists of two recursive neural networks. One network composes the syntactic embeddings of the tokens in the tree, the other composes the semantic embeddings. Both networks implement a pairwise composition operator. For both networks, the order in which embeddings are combined is defined by the structure of the constituency parse tree of the input sentence. A constituency parse tree represents the grammatical structure of a sentence by breaking it down into hierarchical constituents. For example, the constituent parse tree of the sample sentence '*a sentence is parsed*' first breaks down the sentence into two phrases: '*a sentence*' and '*is parsed*'. Then it breaks down the phrase '*a sentence*' into '*a*' and '*sentence*', and the phrase '*is parsed*' into '*is*' and '*parsed*'. To use the parse tree to sequence the pair-wise compositions, we convert the parse tree into a binary tree at the start of the composition process by introducing auxiliary nodes. Then we compose the representation of each node in the binarised parse tree, i.e., words, phrases, and the sentence, by recursively applying the compositional operator from leaves to the root. In the previous sample sentence, first we compose the representations of phrases '*a sentence*' and '*is parsed*' from representations of leaf nodes (i.e., words): '*a*', '*sentence*', '*is*', and '*parsed*'. Then we compose the representation of the sentence from representations of these phrases. Importantly, each composition performed by the semantic composition network is preceded by the corresponding syntactic composition. This is important because the result of the syntactic composition is fed into the semantic network. Consequently, while the syntactic compositional operator ($\oplus$) is a regular operator with learnable parameters, some of the parameters of the semantic compositional operator ($\otimes$) are generated from the syntactic embeddings created from the syntactic compositions implemented by the recursive syntactic network. Generally, hypernetworks

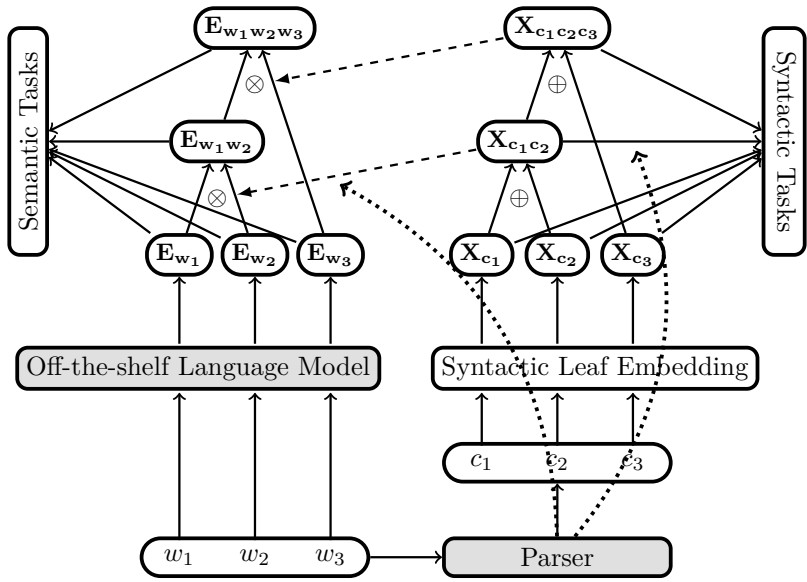

Figure 1: Model Architecture

are deep learning architectures used to generate parameters of a primary network. In this case, the semantic recursive network acts as the primary network, and the syntactic recursive network acts as the corresponding hypernetwork that generates parameters of the primary network (semantic recursive network).

In a regular recursive network, there will be a unique composition operator to generate the representation of any node in the constituency parse tree. However, one can argue that the composition operator to generate the semantic representation of a noun phrase can be different from the compositional operator to generate the semantic representation of a sentence or a verb. The hypernetwork architecture enables the model to learn different semantic compositional operators for different syntactic categories.

We formally define the syntactic compositional operator $\oplus$ as a binary operator on a $dx$-dimensional syntactic embedding space and the semantic compositional operator $\otimes$ as a binary operator on a $dm$-dimensional semantic embedding space.

$$\oplus : \mathbb{R}^{dx} \times \mathbb{R}^{dx} \longrightarrow \mathbb{R}^{dx} \qquad \otimes : \mathbb{R}^{dm} \times \mathbb{R}^{dm} \longrightarrow \mathbb{R}^{dm}$$

Given these operators, the syntactic and semantic representations of a parent node $p$ in a constituency parse tree are generated by the composition of its left child $l$ and its right child $r$,

$$X_p = \oplus(X_l, X_r) \qquad E_p = \otimes(E_l, E_r; X_p)$$

where $X_p$, $X_l$, and $X_r$ are the $dx$ dimensional syntactic embeddings and $E_p$, $E_l$, and $E_r$ are the $dm$ dimensional semantic embeddings of $p$, $l$, and $r$.

As an initial basis for our work, we selected a Tag-Guided Hyper-Tree-LSTM compositional operator proposed by Shen et al. (2020), in which the syntactic compositional operator is a regular Tree-LSTM and the semantic compositional operator is a modified Tree-LSTM in which the parameters are generated by the syntactic embedding. The state of each node in the Tree-LSTM is a tuple $(h, c)$ where $h$ is the hidden state and $c$ is the cell state of the node [3]. The regular Tree-LSTM architecture of the syntactic composition operator ($\oplus$) takes the state of both the left child ($X_l = (\bar{h}_l, \bar{c}_l)$) and the right child ($X_r = (\bar{h}_r, \bar{c}_r)$) from the syntactic recursive network as its input if the node is not a leaf node. If it is a leaf node, then it will take the syntactic leaf embedding ($x$) as its input. These two cases can be combined and considered $X_l$, $X_r$, and

---

[3]We can concatenate both $h$ and $c$ to treat it as a single input vector.

$x$ as the input to the syntactic compositional operator, in which $x$ is set to 0 if the node is not a leaf node, and both $X_l$ and $X_r$ are set to 0 if the node is a leaf node. The syntactic compositional operator outputs the state of the parent ($X_p = (\bar{h}_p, \bar{c}_p)$). In the semantic compositional operator ($\otimes$), along with the state of the left child ($E_l = (h_l, c_l)$), the right child ($E_r = (h_r, c_r)$), and the leaf embedding ($e$), it takes the state of the parent from the syntactic recursive network ($X_p = (\bar{h}_p, \bar{c}_p)$) and outputs the semantic representation of the parent ($E_p = (h_p, c_p)$). Both the syntactic compositional operator and the semantic compositional operator are mathematically described as:

$$\oplus : ((\bar{h}_l, \bar{c}_l), (\bar{h}_r, \bar{c}_r), x) \longrightarrow (\bar{h}_p, \bar{c}_p),$$

$$\bar{h}_p = \sigma(\bar{o}_p) \odot tanh(\bar{c}_p),$$

$$\bar{c}_p = \sigma(\bar{i}_p) \odot tanh(\bar{u}_p) + \sigma(\bar{f}_l) \odot \bar{c}_l + \sigma(\bar{f}_r) \odot \bar{c}_r,$$

$$\bar{i}_p = \bar{V}_{ip}x + \bar{W}^l_{ip}\bar{h}_l + \bar{W}^r_{ip}\bar{h}_r + \bar{b}_{ip},$$

$$\bar{f}_l = \bar{V}_{fl}x + \bar{W}^l_{fl}\bar{h}_l + \bar{W}^r_{fl}\bar{h}_r + \bar{b}_{fl},$$

$$\bar{f}_r = \bar{V}_{fr}x + \bar{W}^l_{fr}\bar{h}_l + \bar{W}^r_{fr}\bar{h}_r + \bar{b}_{fr},$$

$$\bar{o}_p = \bar{V}_{op}x + \bar{W}^l_{op}\bar{h}_l + \bar{W}^r_{op}\bar{h}_r + \bar{b}_{op},$$

$$\bar{u}_p = \bar{V}_{up}x + \bar{W}^l_{up}\bar{h}_l + \bar{W}^r_{up}\bar{h}_r + \bar{b}_{up},$$

$$\otimes : ((h_l, c_l), (h_r, c_r), e; (\bar{h}_p : \bar{c}_p)) \longrightarrow (h_p, c_p)$$

$$h_p = \sigma(o_p) \odot tanh(c_p),$$

$$c_p = \sigma(i_p) \odot tanh(u_p) + \sigma(f_l) \odot c_l + \sigma(f_r) \odot c_r,$$

$$i_p = z^v_{ip} \odot V_{ip}e + z^w_{ip} \odot (W^l_{ip}h_l + W^r_{ip}h_r) + z^b_{ip},$$

$$f_l = z^v_{fl} \odot V_{fl}e + z^w_{fl} \odot (W^l_{fl}h_l + W^r_{fl}h_r) + z^b_{fl},$$

$$f_r = z^v_{fr} \odot V_{fr}e + z^w_{fr} \odot (W^l_{fr}h_l + W^r_{fr}h_r) + z^b_{fr},$$

$$o_p = z^v_{op} \odot V_{op}e + z^w_{op} \odot (W^l_{op}h_l + W^r_{op}h_r) + z^b_{op},$$

$$u_p = z^v_{up} \odot V_{up}e + z^w_{up} \odot (W^l_{up}h_l + W^r_{up}h_r) + z^b_{up},$$

$$z^j_i = U^j_i\bar{h}_p + a^j_i$$

Note that, in the semantic compositional operator, we are using a similar set of parameters as in the syntactic compositional operator and then scaling each output with $z$, and this $z$ is generated from the syntactic embedding of the parent. Through this scaling, we are allowing the model to modify the parameters of the semantic compositional operator depending on the syntactic embedding of the parent.

The model recursively applies the compositional operators on a constituency parse tree of input text, and for that, the embeddings of all leaf nodes are required. These leaf node embeddings can be learned, or we can provide them from off-the-shelf language models. We introduced an embedding layer to learn the embeddings of the syntactic categories of leaf nodes. In the semantic recursive network, we use the embeddings of word tokens generated by off-the-shelf language models such as GloVe, RoBERTa, or Llama. This allows the semantic recursive network to use the linguistic information from existing language models. Then, to generate the representation of any node $p$, we concatenate its syntactic embedding ($X_p$) and semantic embedding ($E_p$), which results in a $dx + dm$ dimensional vector (note that this concatenation of the syntactic and semantics embeddings is additional to the hypernetwork process whereby the syntactic embedding is used as input to the semantic compositional operators, and so the syntactic embedding is used at two levels in our framework: within the hypernetwork process and in this concatentation operation).

## 3.1 Proposed Tasks

We train our model on a set of 6 self-supervised tasks that take the embedding (syntactic or semantic) of any node of the parse tree and output the target label, which is generated from the input itself. Tasks that take syntactic embedding as input and output syntax-related targets (for example, the syntactic category of the node) are called syntactic tasks and tasks which take semantic embedding as input and output semantic targets (for example, generating the string that corresponds to the node) are called semantic tasks. All the proposed 6 tasks are listed below.

1. ***self category prediction (SelfCat)***: Predict the syntactic category (e.g., NP, VP, etc.) of a node from its syntactic embedding. This is to encourage the model to encode the category information in its syntactic embedding.

2. ***neighbour category prediction (SibCat)***: Predict the syntactic category of the sibling of a node by using its syntactic embedding. This is to encourage the model to encode the category information of its sibling in its syntactic embedding.

3. ***syntactic position prediction (SynPos)***: Predict whether the node is a left child or right child from its syntactic embedding. This is because the *neighbour category prediction* task does not use the information about the position of the node, and such position information can be useful. This task will force the model to encode the positional information in its syntactic embedding.

4. ***self text generation (SelfStr)***: Generate the text that corresponds to a node from its semantic embedding. This is similar to an auto-encoder (tree auto-encoding), which forces the semantic embedding of each node to encode the information in the corresponding text. Note that this task is different from the Recursive Auto Encoder used by Socher et al. (2011b), in which the representation of the parent is predicted from the representations of the children. In our case, instead of generating representations, we directly generate or decode the string of the node itself. In this text generation task, phrases in the higher levels of the parse tree will have to predict longer sequences. This can be seen as analogous to the decision to consider a longer context window for higher-level phrases in the CPHRASE model by Pham et al. (2015). To justify this decision, they argue that: *"for shorter phrases, narrower contexts are likely to be most informative (e.g., a modifying adjective for a noun), whereas for longer phrases and sentences it might be better to focus on broader "topical" information spread across larger windows"*. This argument justifies our task, which predicts longer sequences from phrases and sentences from higher levels of the parse tree.

5. ***neighbour text generation (SibStr)***: Generate the text corresponding to the sibling node by using the semantic embeddings of a node. This task is to force the model to encode the information about the left or right context of each node in its semantic embedding.

6. ***semantic position prediction (SemPos)***: Predict whether the node is a left child or a right child from its semantic embedding. This is to encode the positional information in the semantic embedding.

Here, our objective is to learn a general (task-agnostic) representation of sentences, and therefore, we excluded supervised downstream tasks for the training. However, if someone wants to use this framework for a specific downstream task, it would be beneficial to fine-tune the model for that particular downstream task.

The task proposed in the Skipthought model (Kiros et al., 2015) and used by many other succeeding models predicts the next and previous sentences. This resembles our task of predicting the neighbouring context. However, in our case, instead of predicting the entire sentences, we predict neighbours of each word, phrase, and sentence in the constituency-parse tree using a recursive neural network and thereby define our context based on the syntactic information from the parse tree. Our tasks are more syntactic and therefore guide the model to learn more syntactic information. The task proposed by the CPHRASE model (Pham et al., 2015) is also similar to our proposed model tasks because both tasks are to predict (or decode) the context from all levels of the parse tree. However, rather than selecting a fixed context window for each node in different levels of the parse tree, our model uses the neighbouring phrase in the parse tree as the context. This enables the model to make sure that the context has a linguistic boundary and that it is syntactically related to the target phrase. Also, instead of predicting words in a context, we generate the context phrase using a decoder as in the Skipthought model (Kiros et al., 2015). Along with the context phrase of each phrase, we generate the phrase itself, and in this way, we integrate the information within a phrase and information from the context of the phrase when training the compositional operators to generate the representation of the phrase.

We train our model using multiple tasks simultaneously, and therefore, it is a multi-task learning problem. For this multi-task learning, we generate and minimise a single aggregate loss from syntactic and semantic tasks. To generate the aggregate loss, we take a weighted sum of each loss from tasks by using learnable weight parameters. To avoid trivial solutions, we followed Liebel & Körner (2018) and added a regularisation term. The final aggregate loss ($L_{agg}$) function is shown in the equation 1.

$$L_{agg} = \sum_t \big( \frac{L_t}{2c_t^2 + \epsilon} + ln(1 + c_t^2) \big) \tag{1}$$

where $L_t$ is the loss and $c_t$ is the learnable weight parameter corresponding to a task $t$. $\epsilon$ is a small positive real constant to make sure that the denominator will not be 0.

## 4    Training

We experimented with three different off-the-shelf models–GloVe (Pennington et al., 2014), RoBERTa (Liu et al., 2019), and Llama (Touvron et al., 2023)–as a means of initialising the semantic leaf embedding. In all cases, we used a learnable embedding layer for initialising the syntactic leaf embedding. While initialising the semantic leaf embedding by using these three off-the-shelf models, we handled some edge cases that are worth noting. In the case of GloVe, some words may not have a pre-generated GloVe embedding. To handle this, we assign a random 300-dimensional embedding for each of such out-of-vocabulary terms. In the case of RoBERTa and Llama, instead of word embedding, these models generate representations corresponding to each token. Some words may have multiple tokens (note that a token will not span into multiple words), and in such a situation, we took the average of the token embeddings corresponding to each word to generate the leaf embedding.

We fixed the dimension of the syntactic embedding as $64^4$, which resulted in 20K parametered syntactic compositional operators for GloVe-based, RoBERTa-based, and Llama-based models. The dimension of the semantic embedding is fixed as $1024^5$, which resulted in semantic compositional operators with 3.7M parameters on the GloVe-based model, 4.9M parameters on the RoBERTa-based model, and 13.5M on the Llama-based model. For the classification tasks (4 out of 6), we selected a simple multi-layer perceptron model with one 100-dimensional hidden layer and ReLU hidden layer activation. For the two text generation tasks, we used a simple LSTM-based auto-regressive decoder by setting its hidden dimension and token embedding dimension to the same as the model dimension. This model iteratively generates tokens by using the semantic embedding of the node and the embedding of the previous token as input to the LSTM cell.

For training the model, we used 100,000 sentences with 2,280,900 tokens from the Wikipedia dataset[6]. We parsed each sentence using the *benepar* parser (Kitaev & Klein, 2018), one of the best performing state-of-the-art parsers, to generate the constituency parse tree and syntactic categories. In some cases, the parser may generate multiple parse trees from an input text (e.g., if the input text contains multiple sentences). In such cases, we introduced a new category named SS, and joined all the parse trees with this newly introduced category node to make a single tree for each input. We trained the model for 5 epochs with Adam optimiser (Kingma & Ba, 2014) with learning rate scheduling (initial learning rate 0.001) and a batch size of 8. The model training, excluding parsing and preprocessing, finished within 19:30 hours, 20 hours, and 27:45 hours for GloVe-based, RoBERTa-based, and Llama-based models, respectively on a single A100 GPU.

## 5    Analysis of Sentence Representation

We first analysed whether the sentence representation generated by our model encodes more linguistic information, as compared with a number of state-of-the-art sentence encoder models. Probing is one of the standard approaches for investigating whether a representation encodes linguistic information in it. Conneau et al. (2018) proposed a set of 10 probing tasks to test the presence of various linguistic information, and we selected these tasks for our analysis.

The first two probing tasks are *Sentence Length* (*SentLen*), where the task is to predict the number of words in the sentence, and *Word Content* (*WC*), where the task is to predict the presence of 1000 preselected words in the sentence. These two tasks are designed to measure the extent of surface-level information about the sentence encoded by an embedding. The next set of tasks includes *Tree Depth* (*TreeDepth*), and *Top Constituent* (*TC*), and these measure the level of syntactic information encoded by the sentence representation. The target of the *TreeDepth* task is to predict the depth of the parse tree of the sentence, and the target of the *TC* task is to predict the categories of the top constituents (i.e., sequence of categories

---

[4]Shen et al. (2020) used 50 as the dimension of syntactic embedding in their hypernetwork architecture, so we ensured that the dimension we use is greater than 50.

[5]We adopted 1024 as the dimension of the semantic embedding so as to ensure that the embedding has a sufficiently large dimension with respect to the initialised leaf embeddings. By using 1024 dimensions, our semantic embeddings have larger dimensions than the dimensions of GloVe (300) and RoBERTa (768) leaf embeddings. The only exception is for the 4096-dimensional Llama leaf embedding, and in that case, to make the model computationally efficient and to fix the model dimension independent of initialised leaf embeddings, we stick to the same 1024 dimension.

[6]enwiki-20240620-pages-articles-multistream.xml.bz2

of constituents immediately below the root node of the parse tree, e.g., "*ADVP NP VP*"). The final set of tasks–*Past Present* (*PP*), *Subject Number* (*Subj*), *Object Number* (*Obj*), and *Coordination Inversion* (*CI*)– measure the presence of semantic information in the sentence representation. The target for the *PP* task is to predict the tense (past or present) of the main clause of the sentence. *Subj* and *Obj* are to predict the number (*NN* or *NNS*) of the subject and the object of the main clause. The target of the final task, *CI*, is to detect sentences in which the order of clauses is inverted. We excluded two of the Conneau et al. (2018) probing tasks: *Bigram Shift* and *Semantic Odd Man Out* because the datasets of both of these tasks contain 50% non-fluent samples, which can affect the parsing of the sentence. Our sentence representation generation relies on a valid parse tree, and therefore, on both of these datasets, we cannot generate a good-quality sentence representation using our model.

We selected Skip-Thought (Kiros et al., 2015), Infersent (Conneau et al., 2017), and SBERT-WK (Wang & Kuo, 2020), as three well-known baseline sentence representation models from the literature that have shown the best performance. Skip-Thought and Infersent are RNN-based (GRU and BiLSTM) models, and SBERT-WK is a Transformer-based model. In terms of training, Skip-Thought is trained on a self-supervised task (next and previous sentence prediction), Infersent is trained on a supervised task (SNLI), and SBERT-WK is not trained on any extra learning task (it generates sentence representation from BERT representation by doing subspace analysis). To include more recent models, we selected two more baselines: the RoBERTa-based supervised SimCSE model (*sup-simcse-roberta-base*) (Gao et al., 2021) and the most recent and recommended version of the SBERT model (*all-mpnet-base-v2*) (Reimers & Gurevych, 2019).

We first parsed all sample sentences from all the datasets by using *benepar* parser and then generated the sentence representation by using the trained model (representations of the root node, i.e., the concatenation of semantic and syntactic embeddings of the root node). We also generated baseline sentence representations for GloVe, RoBERTa and Llama by taking the average of all semantic leaf embeddings. Then, for each task, we trained a probing model, a simple MLP model with one hidden layer with 100 nodes and ReLU hidden activation function, with corresponding training samples, and we report the accuracy on the corresponding test samples in Table 1.

Table 1: Accuracy on standard probing tasks and their average by excluding *WC*. The best overall scores for each column are shown in bold, and the best scores for each of the three initialised leaf embedding categories (GloVe, RoBERTa and Llama) are underlined.

| | SentLen | TreeDepth | TC | PP | Subj | Obj | CI | WC | Avg |
|---|---|---|---|---|---|---|---|---|---|
| Skip-thought | 0.8603 | 0.4122 | 0.8277 | **0.9005** | 0.8606 | 0.8355 | 0.7189 | 0.7964 | 0.7737 |
| Infersent | 0.8425 | 0.4513 | 0.7814 | 0.8802 | 0.8613 | 0.8231 | 0.7034 | **0.8974** | 0.7633 |
| SBERT-WK | 0.9240 | 0.4540 | 0.7920 | 0.8888 | 0.8645 | 0.8453 | 0.7187 | 0.7750 | 0.7839 |
| SBERT | 0.6043 | 0.2994 | 0.6302 | 0.8730 | 0.7856 | 0.7673 | 0.6141 | 0.6617 | 0.6534 |
| SimCSE | 0.5865 | 0.2889 | 0.5668 | 0.8680 | 0.8157 | 0.7869 | 0.5995 | 0.6077 | 0.6446 |
| Avg GloVe | 0.5936 | 0.3477 | 0.6553 | 0.8427 | 0.7822 | 0.7467 | 0.5468 | 0.8669 | 0.6450 |
| *Ours+GloVe* | **0.9553** | 0.4784 | **0.8756** | 0.8809 | 0.9036 | 0.8794 | 0.6895 | 0.4436 | 0.8090 |
| Avg RoBERTa | 0.8213 | 0.4290 | 0.7496 | 0.8759 | 0.8645 | 0.8417 | 0.7322 | 0.6260 | 0.7592 |
| *Ours+RoBERTa* | 0.9480 | 0.4879 | 0.8712 | 0.8837 | **0.9164** | **0.8867** | 0.7387 | 0.4090 | **0.8189** |
| Avg Llama | 0.7768 | 0.3985 | 0.8223 | 0.8545 | 0.8842 | 0.8482 | **0.7493** | 0.5555 | 0.7620 |
| *Ours+Llama* | 0.9389 | **0.4933** | 0.8746 | 0.8690 | 0.9093 | 0.8856 | 0.7203 | 0.2541 | 0.8130 |

Out of 8 probing tasks considered for this analysis, WC is somewhat an outlier task. First, the performance of most of the recent language model baselines in WC (62% with the RoBERTa average, 61% with the BERT average and 50% with the BERT CLS token) is much worse than other older models and very simple bag-of-words baselines (80% with GloVe average and 98% with tf-idf features). Second, WC has a much larger number of target labels (1000) compared to other tasks. Such a huge number of target labels significantly

increases the number of parameters in the MLP classifier for this task, and this increases the chance of overfitting[7]. For these reasons, we treat WC as an outlier task and we exclude it from our analysis.

Our results, shown in Table 1, are really promising. Adding our model to GloVe, RoBERTa, or Llama results in a significant performance improvement over the corresponding average baseline representations on almost all tasks considered for our analysis. The only exception is the *CI* task with Llama; in this case, Llama with our model did not achieve a better performance than the Llama average. Moreover, our model, in combination with GloVe, RoBERTa or Llama, outperforms other sentence representation models in the literature on all tasks except *PP*. Interestingly, even in combination with a non-neural word embedding model, GloVe, our model achieves better performance than the RoBERTa and Llama averaged baselines and other neural state-of-the-art sentence representation models such as SBERT-WK (Wang & Kuo, 2020) on 5 of the 7 tasks (*SentLen*, *TreeDepth*, *TC*, *Subj*, *Obj*). When we take the average accuracy across all the tasks by excluding the outlier *WC* (see the last column of Table 1), we find that our model with GloVe, RoBERTa, and Llama achieves better average scores than the corresponding average baselines and state-of-the-art sentence representation models. This indicates that the sentence representation generated by our model encodes more linguistic information than baseline representations and other state-of-the-art representations.

### 5.1 Ablation Study: Impact of Self-supervised Tasks

To study the impact of the six proposed self-supervised tasks, we conducted an ablation study by systematically excluding some of these tasks from model training and then analysing how much performance drop this resulted in. To measure the performance drop, we evaluated on 7 of the probing tasks, excluding the WC, the outlier task on which our model is not performing well. Then we reported the average accuracy drop as the measure of the impact of the excluded self-supervised task(s).

In the first set of ablation studies, we dropped each task in turn and measured the impact of each of the individual self-supervised tasks. The performance drops corresponding to each of the 6 self-supervised tasks are shown in the first 6 rows of Table 2. Our self-supervised tasks can be grouped into two categories: 1. syntactic tasks, i.e., tasks that are based on the syntactic representation (*SynPos*, *SelfCat*, and *SibCat*) and 2. semantic tasks, i.e., tasks that are based on semantic representations (*SemPos*, *SelfStr*, and *SibStr*). The impact of the syntactic tasks as a whole on learning sentence representations is an interesting question, and to address this question, we did an ablation study by dropping 3 syntactic tasks (*SynPos*, *SelfCat*, and *SibCat*)[8] and reported the corresponding performance drop in the 7th row of Table 2.

Table 2: Results of the self-supervised task ablation study: the average (across 7 probing tasks, excluding WC) drop in performance when ablating learning task(s) (positive value indicates drop)

| Task Ablation | GloVe | RoBERTa | Llama |
|---|---|---|---|
| $-SynPos$ | 0.000 | 0.002 | $-0.002$ |
| $-SelfCat$ | $-0.002$ | 0.004 | $-0.006$ |
| $-SibCat$ | $-0.001$ | 0.005 | $-0.004$ |
| $-SemPos$ | 0.000 | 0.001 | $-0.005$ |
| $-SelfStr$ | **0.088** | **0.088** | **0.101** |
| $-SibStr$ | $-0.001$ | 0.010 | 0.002 |
| $-Syn$ | 0.005 | 0.008 | $-0.002$ |

From our ablation study, we found a significant performance drop in all models (8.8% for GloVe-based and RoBERTa-based models and 10.1% for the Llama-based model) after excluding the self text generation task

---

[7]Note that, for this evaluation, we trained an MLP classifier for each of the probing tasks in a supervised fashion by using the standard probing task dataset. Such supervised learning with a large number of target labels increases the chance of overfitting.

[8]One can think about dropping the semantic task to measure the impact of semantic tasks on our model. However, if we drop semantic tasks, then the semantic compositional operator, and therefore the entire semantic network, will not be able to learn. This will result in a situation where the information in the initialised leaf embedding from the existing language model will not be used in the final representation. In that situation, the final representation will encode only the category information of the root node, and most likely, this will result in an obvious and drastic performance drop. Such an ablation study will be less interesting, and therefore, we didn't include it.

(*SelfStr*) from the set of 6 self-supervised tasks. This shows the crucial role of self text generation task for training our model. We found that, individually, the other tasks are not as crucial as the self text generation task.

For the RoBERTa-based model, we found performance drops by ablating each of the 6 individual self-supervised tasks, and this suggests that all tasks are useful for the training of the RoBERTa-based model. However, surprisingly, individually dropping some of the other self-supervised tasks from the training of GloVe-based and Llama-based models resulted in a very small performance improvement (less than 1%). One possible reason is that dropping less useful tasks may enable the model to focus more on other useful tasks, and this can result in a performance improvement. However, given the positive impact of all of these tasks on the RoBERTa-based model, it is not advisable to drop any of these three tasks for the training of our model.

Our ablation study on the syntactic task set shows a performance drop of 0.5% and 0.8% for the GloVe-based and RoBERTa-based models, respectively. These results indicate that, while the contribution of syntactic tasks is not highly significant, they are still useful to some extent for these two models. Notably, in the GloVe-based model, individual syntactic tasks do not appear to contribute, but they have a positive effect when used in combination. In contrast, for the Llama-based model, we did not observe any noticeable performance degradation, suggesting that syntactic tasks are not particularly beneficial for this model variant.

Although syntactic tasks have a limited impact on overall model performance, this does not undermine our claim regarding the explicit integration of syntactic information into the framework. This is because the semantic tasks also contribute to learning both the semantic and syntactic recursive networks. As a result, even in the absence of syntactic tasks, the model still learns syntactic information within the syntactic recursive network.

## 6 Analysis of Compositional Operator

In our model, we are applying the compositional operator recursively to generate the representations of phrases and sentences. The cumulative effect of this recursive composition may lead to the case where some of the properties of the sentence representation can vanish or explode if the constituency parse tree is deep. To evaluate how robust our model is to such pitfalls, we measured some properties of representations of nodes in the constituency parse tree. Then we analysed how these properties vary with varying heights of the node. Note that the height of a node is the same as the number of times the compositional operator is applied to generate the node's representation, and therefore, if some properties are decreasing or increasing rapidly with the increasing height, this indicates that the property is likely to vanish or explode for deeper constituency trees.

One of the interesting properties of vector representations is their isotropy. Isotropy is the measure of how uniformly the representation vectors are distributed in the embedding space, and it indicates how well the model utilises the embedding space. If we apply composition recursively, the generated representations can become more and more skewed and shrink to smaller and smaller representational subspaces. If the set of all representations lies in a smaller subspace of the embedding space or if the distribution is skewed in a particular direction, then this limits the model's capacity to encode information because the effective dimensionality of its embedding space will be reduced. Different approaches to measuring the isotropy of representations are used in the literature (Mu et al., 2018; Cai et al., 2021; Rudman et al., 2022). Of these approaches, we use the measure called IsoScore proposed by Rudman et al. (2022) because it has become relatively standard in the literature and has the properties of being mean agnostic and rotation invariant. The IsoScore metric returns values between 0 and 1, and a high IsoScore for a set indicates that the elements of the set are relatively uniformly distributed. In the context of this work, a high IsoScore indicates a good utilisation of an embedding space by an embedding model. Another interesting property of vector representations is the norm of the generated representation. If we apply composition recursively, one potential risk is that the composed representations shrink to the origin or explode.

We analysed both the isotropy and the norm of representations of nodes (semantic embedding concatenated with syntactic embedding) with varying height. For the analysis, we selected 10,000 sample sentences from

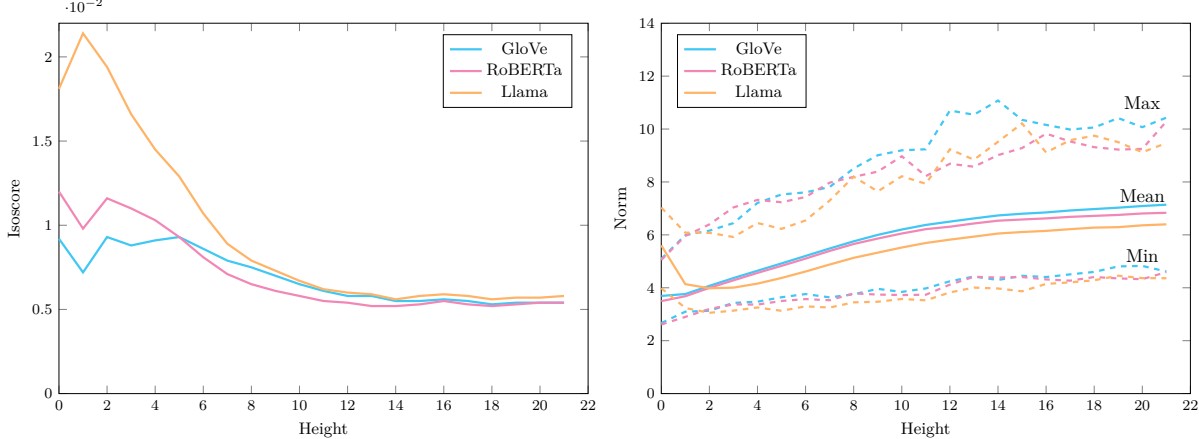

Figure 2: IsoScores and Norms across varying height of nodes

the Wikipedia dataset. To avoid any possible bias of our model towards the dataset used for training, we ensured that none of the samples used for this analysis were from the data used to train our models. The norm (minimum, maximum and mean values) and the isotropy of nodes with varying height are presented in Figure 2. Although there is a slight trend of increasing norm length and decreasing isotropy during the initial compositions, both properties stabilise for higher values of height. This rules out the possible issue of vanishing or exploding values of norm or isotropy on longer sentences (or deeper parse trees).

## 6.1 Sensitivity of Semantic Composition on Syntactic Embedding

An interesting question is to what extent the semantic composition operation is affected by the syntactic information fed into it via the hypernetwork architecture. To analyse this, we measure the sensitivity of the semantic compositional operator to changes in the syntactic embeddings. If the operator is not sensitive to changes in its syntactic embedding, this would indicate that the hypernetwork architecture does not affect the semantic composition and vice versa.

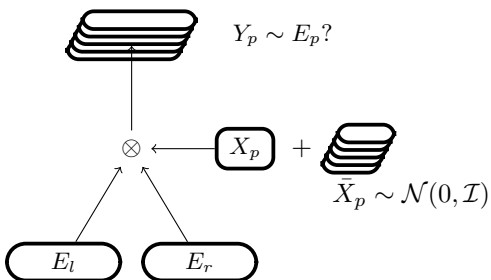

Figure 3: Sensitivity of Semantic Compositional Operator on Syntactic Embedding

To measure the sensitivity of the learned semantic compositional operator to the syntactic embedding it receives as input, we systematically altered the syntactic embedding ($X_p$) and measured how much change is present in the output of the semantic operator in comparison with its normal output ($E_p$) when the syntactic embedding is not modified. Our framework for this analysis is shown in Figure 3. The change in the output can be measured either in terms of isotropy (whether these output variants are distributed uniformly across all dimensions?) or in terms of its spread (how close are these output variants to the normal output $E_p$?). To measure the isotropy, we used the method of IsoScore proposed by Rudman et al. (2022), and to measure the spread, we calculated the average Euclidean distance from the normal output.

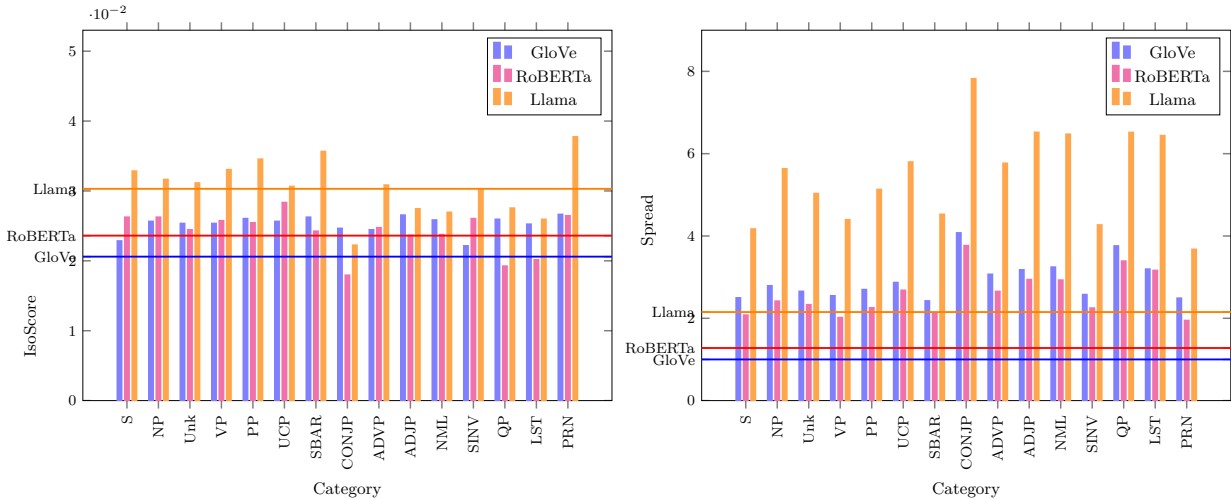

Figure 4: Category-wise IsoScore and Spread of Semantic Compositional Operator to Changes in Syntactic Embedding

For this analysis, we selected all non-leaf nodes from 100 sample sentences from the Wikipedia dataset (ensuring that the selected samples were not used for the training). Then, for each node, we generated 10,000 altered variants of its syntactic embedding $(X_p)$ by adding a random embedding $(\bar{X}_p)$ sampled from a multivariate normal distribution with zero mean and identity standard deviation. Then we generated 10,000 semantic embeddings of the node $(Y_p)$ from the semantic embeddings of its left child $(E_l)$ and right child $(E_r)$ by using the semantic compositional operator $(\otimes)$ with each of the 10,000 altered variants of the syntactic embedding $(X_p + \bar{X}_p)$. Then the IsoScore and spread are calculated from these 10,000 altered variants of the semantic embeddings of the node $(Y_p)$. We averaged the IsoScore and spread of nodes according to the syntactic category of the node.

To get an idea about the goodness of these IsoScores and spreads, and in turn the sensitivity of the semantic compositional operator to syntactic embedding, we calculated the baseline IsoScore and the baseline spread values from semantic embeddings (non-perturbed) of all the nodes used for this analysis (a total of 2162 non-leaf nodes from the selected 100 Wikipedia sample sentences). The spread of semantic embeddings is calculated by taking the average Euclidean distance from its centroid. If the average IsoScore or spread of a category is greater than (or at least comparable to) these baseline scores, then that indicates that, for that category, the semantic compositional operator is sensitive to the changes in the syntactic embedding.

The average IsoScores and spreads of the semantic nodes broken down by their corresponding syntactic categories, along with the baseline values, are plotted in Figure 4. We found that, for all three base-model variants, i.e., GloVe-based, RoBERTa-based, and Llama-based compositional operators, the average spreads of values of all syntactic categories are appreciably larger than the baseline spread, and the average IsoScores of almost all categories are larger than or comparable with the baseline IsoScore. This indicates that, regardless of the syntactic category, the semantic compositional operator is sensitive to changes in syntactic embeddings.

From our analysis, we observed that the semantic composition operator has varying sensitivity with respect to the syntactic embedding across different syntactic categories. A possible interpretation of these results is that, for less frequent categories, the semantic compositional operator can have a lower sensitivity to the syntactic embedding due to insufficient training. To investigate this possibility, we calculated the frequencies of syntactic categories in the training data and calculated their correlations with the IsoScores and spreads. We found that the Pearson correlation coefficients of syntactic category frequency and IsoScore are 0.0593, 0.2654, and 0.2244 for GloVe-based, RoBERTa-based, and Llama-based compositional operators, respectively. These positive syntactic frequencies by IsoScore correlation scores indicate that, in general, the semantic composition operator is less sensitive to less frequent syntactic embeddings in terms of IsoScore.

Although for the GloVe-based semantic operator, this variation in sensitivity with respect to the frequency of a syntactic category is relatively weak, for both the RoBERTa-based and Llama-based operators, this phenomenon is relatively strong. The Pearson correlation coefficients of the spread of embeddings with the syntactic category frequency are $-0.3131$, $-0.3284$, and $-0.1809$ on GloVe-based, RoBERTa-based, and Llama-based semantic compositional operators, respectively. These negative correlations imply that the less frequent categories have a relatively high spread, even though they have a relatively low IsoScore. In summary, even if the altered variants of the semantic embedding of less frequent categories lie in a smaller subspace, they disperse more across this space. This implies that even for an extremely infrequent category, the semantic compositional operator does not become insensitive to the syntactic embedding.

## 7 Conclusion and Future Work

We proposed a self-supervised recursive hypernetwork architecture to generate sentence representations that encode more linguistic information within the generated embeddings. We validated that the generated sentence representations encode more linguistic information than the standard average-based baselines and state-of-the-art sentence representation models. Our ablation study validated the impact of each of the proposed self-supervised tasks on our model training, and our analysis of the composition operator verified the stability of our framework with increasing length of sentence (more precisely, the depth of the parse tree). We also observed that the hypernetwork architecture has an impact on our framework, and our semantic compositional operator adapts depending on the syntactic category of the node.

Our approach for generating sentence representations has a broader impact on the performance of sentence-level tasks, because generating a representation with more linguistic information will likely be useful for downstream tasks. Also, in that case, fine-tuning our framework for specific downstream tasks will likely result in further performance improvement. More importantly, our framework can be used with any language model, and such scalability opens up the potential benefit of our framework with more recent language models for generating better sentence representations that can lead to better performance.

Compared to pretrained language models, our framework is generally lightweight with a few million parameters, and can be trained with a relatively small dataset (we trained the model with 100,000 sentences). Yet, we observed that our model, even with a relatively simple GloVe representation, encodes more linguistic information than both of the baseline averaged RoBERTa-based and Llama-based representations. This indicates that our model has implications from an efficiency perspective.

One of the limitations of the recursive neural network is its dependence on the constituency parse tree of each input. Sachan et al. (2021) observe that there is a significant performance improvement by using gold standard parse trees instead of trees generated by the off-the-shelf stanza parser. In our model, we used another off-the-shelf parser, called *benepar*, to generate parse trees. To verify the dependence of our model towards the quality of the generated parse tree by using *benepar* parser, we evaluated our model on the *Bigram Shift* and *Semantic Odd Man Out* task from the standard probing task. 50% of the samples in the datasets of these tasks contain non-fluent sentences, and on such samples, the parse tree generated by our parser is expected to be worse. We found that on both of these tasks, our model's performance is not good, and that shows the dependence of our model on good-quality parse trees. In future work, we will systematically explore the sensitivity of the framework to parser performance.

Another limitation of our work is that we only experimented with our model on the English language. Our framework can be applied to any language; however, the requirement of a good-quality parser can be a major limitation. Maillard et al. (2017) and Hu et al. (2022) learned the parse tree along with the sentence representation by considering all possible branchings of the parse tree and selecting the best branching to apply composition. Expanding our experiment to other languages by adopting such techniques to overcome the requirement of the parser is also proposed for future work.

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
