# OpenReview forum: "Know Yourself and Know Your Neighbour : A Syntactically Informed Self-Supervised Compositional Sentence Representation Learning Framework using a Recursive Hypernetwork"
_TMLR — Accepted by TMLR_

### Review · Reviewer_kki5 · 2025-07-14

**Summary Of Contributions:**

In their paper "Know yourself and know your neighbor: a syntactically informed self-supervised compositional sentence representation learning framework using a recursive hypernetwork", the authors suggest an approach for self-supervised training of sentence representations based on parsing trees. The method recursively aggregates word representations over the parsing tree, and is trained to produce meaningful aggregations using several syntactic tasks.

Strengths: Interesting approach, and quite novel (at least when applied to BERT embeddings) as far as I am aware. Some promising evaluation results.

Weaknesses: Evaluation results are not super strong (but that is okay for TMLR). Some obvious ablations are missing. Stronger SOTA comparisons are missing.

Overall, I think this could be published in TMLR if authors add some additional comparisons and ablations.

**Audience:**

Yes

**Audience Explanation:**

I think the paper explores an interesting avenue, and while some of the results are a bit mixed, this is an interesting contribution.

**Broader Impact Concerns:**

No concerns.

**Claims And Evidence:**

Yes

**Claims Explanation:**

Yes, but I would like to see some more comparisons and some ablations.

**Requested Changes:**

MAJOR COMMENTS

* All comparison models used in section 5 are from 2020 or earlier. I am missing a more recent comparison model. For example, SBERT is very popular, and though the preprint is from 2019, their current recommended all-purpose model is all-mpnet-base-v2 and it's more recent and performs well in MTEB. At the minimum, I would like the authors to evaluate all-mpnet-base-v2. They can also optionally add some other more recent models with good MTEB performance.

* There are no ablations. How would your models perform if you entirely remove syntactic component, and use identical fine-tuning procedure with only semantic embeddings? Alternatively, how would your models perform if you only have syntactic embeddings and entirely remove semantic ones? I think that could be interesting experiments to add.


MEDIUM COMMENTS

* The abstract is really short and does not contain any specific details. It can easily be twice as long and provide some more details.

* Some terms are never properly defined. What is a "recursive neural network"? What is a "hypernetwork"? These are not standard terms, please define them. Also, what exactly is a "constituency parse tree"? Not all readers will know that. Can you give an example tree for an example sentence?

* RoBERTa generates embeddings of tokens, not words. And tokens will not always align with words. But your parse trees presumably have words as leaves. How do you deal with that? This is never explained, or maybe I missed it.

* In the literature "sentence embedding" often refers to an embedding for the whole input text which can consist of multiple sentences (e.g. in SBERT). How do you handle that with your parse trees? This needs to be clarified.

* Can you add "Average" column to Table 1 which would average over 10 tasks?

* Why not use MTEB for evaluation? Please comment somewhere on how your probing tasks differ from MTEB.


MINOR COMMENTS

* \citet is often used instead of \citet

* "supervised methods" are mentioned on page 1 but not really defined. Please define what you mean by "supervised" in this context. Also in Section 2.2. Models like SBERT are pre-trained in self-supervised way and then finetuned in a supervised way (meaning that they use datasets of paired text like question/answer pairs for contrastive training). This should be clearly explained.

* page 2: "recursive hypernetwork trained" -- sentence ends too early.

* "hypernetwork" is sometimes written with a hyphen and sometimes without.

* page 9: "increase the change of overfitting" -- this is unclear. What overfitting? You are training in a self-supervised way, why would there be an overfitting?

* page 12: negative numbers should be typeset in mathmode, e.g. $-5$ and not -5.

---

### Review · Reviewer_VGid · 2025-08-12

**Summary Of Contributions:**

The paper introduces a framework for learning sentence representations that makes use of syntactic information. It uses two recursive networks: one builds syntactic embeddings, and the other builds semantic embeddings, with the syntactic part helping to set parameters for the semantic part. Six training tasks are used to encourage the model to capture both meaning and structure. The system is trained on about 100k Wikipedia sentences and tested on ten probing tasks, showing better results than simple averaging of GloVe or RoBERTa embeddings and outperforming several older methods (Skip-Thought, InferSent, SBERT-WK). The authors also examine how embedding properties change with parse-tree depth and how sensitive the semantic composition is to changes in syntactic input.

**Audience:**

Yes

**Audience Explanation:**

This paper relates to severl themes: sentence representation learning, syntax injection and self‑supervised learning, which are of interest to the TMLR community. The proposed combination of a recursive hyper‑network with multiple self‑supervised tasks is novel and could inspire follow‑up work. However, without stronger empirical validation, it is difficult to recommend adoption.

**Broader Impact Concerns:**

None.

**Claims And Evidence:**

No

**Claims Explanation:**

The idea of a hyper‑network that conditions semantic composition on syntactic embeddings and the formulation of several self‑supervised tasks are interesting. However, the empirical evidence provided in the paper is not entirely convincing.

- The model is trained on only 100k Wikipedia sentences, which is a small fraction of modern pretraining corpora. It is unclear whether the reported gains would generalise to larger datasets or to other domains.

- The main comparison is against GloVe and RoBERTa embeddings and three relatively old models (Skip‑Thought, InferSent, SBERT‑WK). Many strong recent baselines for sentence representation learning (e.g., SimCSE) are absent.

- Moreover, improvements are not consistent: in Table 1, while several tasks see notable boosts (e.g., SentLen, TC, Subj), others degrade substantially (e.g., WC drops from 0.8669 to 0.4436 for GloVe, and from 0.6260 to 0.4090 for RoBERTa).

**Requested Changes:**

- Include stronger baselines and verify the proposed model on a larger-scale dataset.

- More in-depth elaboration of inconsistent improvement in Table 1 (It is also recommended to add an average column if applied.).

- Ablation study: assess the contribution of each self‑supervised task and the importance of the hyper‑network.

---

> ### Author Response · Authors · 2025-08-18
>
> Thank you for your comments and suggestions to improve the strength of the empirical validation of our work. Please find our response to your comments below:
>
> The reviewer highlighted an inconsistent improvement of our models in Table 1. However, as we can see in the paper, the performance drops are showing only on 3 tasks, i.e., Word Content, Bigram Shift, and SOMO, and we think these 3 cases are outliers. Both the performance drop from the Bigram Shift and the SOMO tasks are an expected behaviour because both datasets contain 50% grammatically incorrect or incoherent samples. Such samples affect the parsing quality, which is the essential prerequisite for our model. So the reason behind the odd behaviour from Bigram Shift and SOMO is attributed to the characteristics of the datasets, not to the objective of the task. Given these facts, the performances of these tasks are not a reflection of our model’s general capability to encode linguistic information, and therefore, we consider these two tasks to be outliers.
> In the case of Word Content, the task is a special case, because the number of target labels is 1000, much higher than all other tasks. We would like to mention here that the reported performances of the Word Content task by using BERT average (61.11%) and by using BERT CLS token (50.15%) are much lower than GloVe (80.61%) [Wang and Kuo, 2020 (SBERT-WK)], which is somewhat an odd behaviour. In our preliminary experiments, we observed that the performance of Word Content varies significantly with the number of hidden neurons and the choice of activation functions in the MLP. Given the sensitivity of the Word Content task towards the architecture of the prediction model and the odd behaviour reported in the literature, we think Word Content is also an outlier task. We admit that we didn’t include sufficient discussion about this matter in the paper, and we are happy to expand the discussion about this issue in the revised paper. Regarding the comment on adding an average column to the Table, we appreciate the suggestion and will include appropriate central tendency measures that take into consideration outlier cases in the revised paper.
>
> Regarding your comments about extending the work on a large-scale dataset, we appreciate your comment, and it is an interesting direction for the extension of our work. However, we would like to clarify that while the model’s behaviour on a larger dataset may offer additional insights, it does not undermine the validity of our current evaluation results. We agree that, if our model couldn’t encode sufficient linguistic information by training with 100K samples, then extending the training dataset with more samples could be helpful for validating our proposed model. However, given the fact that our model obtained better performance on all tasks except 3 extreme cases (Word Content, Bigram Shift, and SOMO), we would like to know how the extension of training with a large dataset will help strengthen the empirical validation. We would also like to highlight that the probing task dataset, which we used for our evaluation, is from the Toronto Book corpus, and our training data is from the Wikipedia dataset. Given this fact, we are not clear in what way the training of our model with a dataset from another domain will help to strengthen the empirical validation. If there is a convincing advantage for extending the training with more datasets, we are happy to do that and report the results; otherwise, given the extra computational budget and time required for this extension, we would like to propose it for future work.
>
> Regarding the comment about including a stronger and recent baseline, we are happy to include more recent models in the updated paper. Based on suggestions, we have already evaluated more recent baselines, including SimCSE and all-mpnet-base-v2. Our results indicate that our approach, combined with both RoBERTa and GloVe, outperforms these new baselines on all tasks except Word Content, SOMO and BigramShift (tasks which we consider as outlier tasks). We will include these new baselines in the revised version.
>
> Thank you for the suggestion to add an ablation study, and we agree that such a study will strengthen the empirical validation. We have performed ablation studies on different self-supervised tasks. In the first set of ablation studies, we dropped each task in turn. Overall, we find that all tasks are useful, with the “self text generation” task being the most important (ablating this task results in a drop of 8.8% in performance for both the GloVe and RoBERTa-based versions of our architecture). We also performed an ablation where we dropped all of the syntactic tasks as a group, and again, we found a drop in performance for both the GloVe and RoBERTa-based versions of our architecture. We are happy to include this ablation study in the revised paper.

---

### Review · Reviewer_eiS2 · 2025-08-24

**Summary Of Contributions:**

Summary: The paper proposes to combine semantic level and syntactic level information together in a dual recursive neural network formulation. It introduces self-supervised learning losses to refine the representations. The resulting network shows good performance on a set of downstream evaluation tasks on natural language involving surface information extraction, tree recognition, and clause recognition, etc., especially on those tasks that involve syntactic understanding.

Strengths:
- Using both syntactic and semantic networks jointly is an interesting idea.
- Using self-supervised tasks to train both semantic and syntactic embedding networks is a novel contribution.

Weaknesses:
- Variety of parsers: The paper only used benepar parser in the experiments. It would be good to try other parsers and see the dependence on the quality of the initial parse.
- Operators defined on the top of Page 5 seem confusing and too general. Since the paper takes a very concrete implementation of an LSTM, it would be better to define it upfront in the paper to avoid confusion.
- Lack of large scale LLMs: The design makes sense if the LLM (e.g. RoBERTa) is shallow, incapable or unaware of the syntactic structure. It is unclear if it can help larger LLMs. It might still be a good story if the target is to help small LLMs, but the sensitivity of the proposed architecture in terms of base LLM scale. What if you extract the embedding from a larger LLM? Given today’s progress in LLMs, it seems necessary to show experiments using larger scale models.
- Ablation of individual components and baselines: While the authors show that the performance is sensitive to the syntactic network, it would be good to show simpler baselines. E.g. only the semantic network with the SSL loss, other composition operators using addition/LSTM/Transformer etc. From the title, it seems the core contribution is the SSL loss, and it would also be good to ablate other loss variants.
- The results seem mixed. While there are a few tests in which the proposed method is better, there is not a clear winner.

**Audience:**

Yes

**Audience Explanation:**

Although the semantic + syntactic design can be less practical in today's mainstream implementations, I think some TMLR audience can be interested in knowing the findings.

**Claims And Evidence:**

No

**Claims Explanation:**

I believe the submission introduces a new design. However, as listed above, I think it can benefit from more clear experimental design that dissects different model components and compares to more baselines.

**Requested Changes:**

- Add more ablation and baselines.
- Run results with larger scale LLMs.
- Add results with different parsers.
- Improve clarity of math notations.

---

> ### Author Response · Authors · 2025-09-04
>
> Thank you for your suggestions to include more ablation studies and baselines, and we will include them in the revised paper. In fact, we have already evaluated more recent baselines, based on suggestions from other reviewers, including SimCSE (SimCSE: Simple Contrastive Learning of Sentence Embeddings https://arxiv.org/abs/2104.08821, EMNLP 2021) and all-mpnet-base-v2 (which is an all-purpose model based on SBERT (ACL 2019) and is the model that is now recommended for use by the SBERT developers). Our results indicate that our approach, combined with both RoBERTa and GloVe, outperforms these new baselines on all tasks except Word Content, SOMO and BigramShift (tasks which we consider as outlier tasks).
>
> We have also performed ablation studies on the SSL tasks. In the first set of ablation studies, we dropped each task in turn. Overall, we find that all tasks are useful, with the “self text generation” task being the most important (ablating this task results in a drop of 8.8% in performance for both the GloVe and RoBERTa-based versions of our architecture). We also performed an ablation where we dropped all of the syntactic tasks as a group, and again, we found a drop in performance for both the GloVe and RoBERTa-based versions of our architecture.
>
> Regarding your suggestion to include more recent large-scale LLMs, we agree that it is interesting to examine how our approach might extend beyond GloVe and RoBERTa. However, most models introduced after RoBERTa are autoregressive Transformer decoders designed for text generation. Such models are generally considered less suitable for sentence representation learning without substantial adaptation, which is why we did not include them in our experiments. However, based on your suggestion, we evaluated the Llama-3.1-8B model and observed that its performance is slightly better than RoBERTa on some tasks and slightly worse on others. When we trained our model on top of Llama, we got a performance improvement compared to the Llama baseline. This suggests that our model is applicable to recent LLMs. However, on all tasks, this Llama-based model is outperformed by our existing RoBERTa-based model and/or our GloVe-based model. Should you consider this result essential to substantiate our claim, we will be happy to include it in the revised paper.
>
> Regarding your comment on using other parsers, we agree that the performance of our approach is likely dependent on the quality of the initial parse, and in the conclusion section of the paper, we note this as one of the limitations of our approach.  In fact, the results of the Bshift task, where the input sentences are ill-formed, provide some empirical evidence that the approach is sensitive to the quality of the parser. For this reason, we selected one of the best-performing constituency parsers—Benepar (reported F1 score of 95.13% on WSJ test set)—for our model, and we will justify this choice in the revised paper. We believe that the results we have obtained using the Benepar parser already support our claim that our approach improves the encoding of linguistic information within the generated sentence representations, and therefore, demonstrating performance with a better parser would not qualitatively strengthen the claim. We also believe that using a poor-quality parser to show the dependence of our model on parser quality is not directly relevant to the claim we make in the paper; although we agree that exploring the interaction between parser performance and our approach is interesting, and we have proposed this as future work. Given this, we are uncertain whether repeating our experiments with another parser would meaningfully strengthen the paper.
>
> Thank you for your suggestion to reduce the chance of confusion about the compositional operator. We will rectify it in the revised paper.

---

### Decision · Action_Editor_1uZr · 2025-10-13

**Recommendation:** Accept with minor revision

**Additional Comments:**

This paper proposes a method for self-supervised sentence representation learning using constituency parse trees. Their method employs recursive networks to compose word embeddings into syntactic and semantic sentence representations, separately, following an ordering determined by the parse tree of the input sentence. The network that composes semantic embeddings is informed by the syntactic recursive network. They train this system on a mixture of 6 self-supervised tasks, for the purpose of producing meaningful sentence representations. The authors evaluate their method on several downstream evaluation tasks on natural language including surface information extraction, tree recognition, clause recognition, etc. The proposed method performs well on several tasks, especially those that require syntactic understanding.

All three reviewers found the method proposed interesting and novel. For instance, Reviewer eiS2 noted that "using both syntactic and semantic networks jointly is an interesting idea" and a "novel contribution"; Reviewer VGid found that using syntactic information to inform semantic composition is "interesting" and "novel".

Reviewers also pointed out the following key weaknesses: (i) missing ablations of individual components; (ii) missing stronger / more recent baselines, (iii) missing experiments on larger-scale models, where the gain from explicitly incorporating syntactic information may be smaller; (iv) the results are mixed, with insufficient discussion. Other, smaller issues, were also pointed out, about the dependence on the specific chosen parser for instance as well as some clarity issues and missing experimental details.

During the rebuttal, the authors have revised their paper to take all of this feedback into consideration. They have ran ablations w.r.t the chosen self-supervised tasks used for training, dropping each task individually, or all syntactic tasks as a group. They have also added comparisons to more recent baselines including including SimCSE and all-mpnet-base-v2.
The authors also experimented with Llama-3.1-8B, finding that building their model on top of plain Llama leads to improving it.
Other comments were also addressed to the satisfaction of reviewers, through adding additional discussions and clarifications to the paper.

Changes required in minor revision:

- Tweak claims like "We validate that the generated sentence representation encodes richer linguistic information than both averaging baselines and state-of-the-art alternatives" to align them with the empirical evidence (which shows mixed results). For example, "on N out of M tasks, we find that the generated sentence representation [...]". Please also include the discussion provided during the rebuttal about the evaluation tasks, and how some of them may be considered as outliers.

- Add all ablations and refine claims accordingly. If removing the syntactic tasks does not make a big difference in the results, remove or modify claims that the inclusion of syntactic SSL training tasks is one of the reasons that contributes to strong performance on tasks that require syntactic understanding. As Reviewer kki5 pointed out, the approach can still be regarded as syntactically informing sentence representations, due to the architecture design and use of parse trees. But the claim that "the self-supervised tasks are designed to guide the compositional operators to encode syntactic information [...]" needs tweaking as the syntactic-aware nature of the generated sentence representations does not seem to be due to the selection of the SSL tasks.

- To substantiate the claim that the proposed method is compatible with larger models, please add the Llama results mentioned in the rebuttal to the paper as well.

- Similarly, please add the results including comparison with more recent baselines to substantiate claims about outperforming recent methods / "state-of-the-art".

**Audience:**

Yes

**Audience Explanation:**

Sentence representation learning is a topic of interest to TMLR and generally valuable, as it is relevant for various downstream tasks. The proposed method performs strongly on certain tasks, especially when syntactic understanding is needed, compared to baselines, and would be of interest to this community.

**Claims And Evidence:**

Yes

**Claims Explanation:**

This paper proposes a method for self-supervised sentence representation learning using constituency parse trees. Their method employs recursive networks to compose word embeddings into syntactic and semantic sentence representations, separately, following an ordering determined by the parse tree of the input sentence. The network that composes semantic embeddings is informed by the syntactic recursive network. They train this system on a mixture of 6 self-supervised tasks, for the purpose of producing meaningful sentence representations. The authors evaluate their method on several downstream evaluation tasks on natural language including surface information extraction, tree recognition, clause recognition, etc. The proposed method performs well on several tasks, especially those that require syntactic understanding.

All reviewers pointed out that important ablations are missing, which makes it challenging to assess the gains produced by (different components of) the proposed method, and in turn makes it challenging to substantiate specific claims about key design choices e.g. about the role of syntactically informing semantic embeddings, etc.
However, the authors have addressed this issue comprehensively during the rebuttal. Please see the "Additional Comments" section for requested minor revisions.

Another issue pointed out by the reviewers is that the results are mixed: while the proposed approach performs well on some tasks (especially those involving syntactic understanding), it does not perform well on all tasks. Similarly, see the "Additional Comments" section for a request for refining claims accordingly.

The above issues can be addressed with minor revisions, as outlined below.